# DGraph: A Large-Scale Financial Dataset for Graph Anomaly Detection

**Xuanwen Huang**[†], **Yang Yang**[†✉], **Yang Wang**[‡], **Chunping Wang**[‡],
**Zhisheng Zhang**[†], **Jiarong Xu**[§], **Lei Chen**[‡], **Michalis Vazirgiannis**[††]

[†]Zhejiang University, [‡]Finvolution Group
[§]Fudan University, [††]École Polytechnique
{xwhuang, yangya, zhangzhsh6}@zju.edu.cn
{wangyang09, wangchunping02, chenlei04}@xinye.com
jiarongxu@fudan.edu.cn, mvazirg@lix.polytechnique.fr

## Abstract

Graph Anomaly Detection (GAD) has recently become a hot research spot due to its practicability and theoretical value. Since GAD emphasizes the application and the rarity of anomalous samples, enriching the varieties of its datasets is fundamental. Thus, this paper present *DGraph*, a real-world dynamic graph in the finance domain. *DGraph* overcomes many limitations of current GAD datasets. It contains about 3M nodes, 4M dynamic edges, and 1M ground-truth nodes. We provide a comprehensive observation of *DGraph*, revealing that anomalous nodes and normal nodes generally have different structures, neighbor distribution, and temporal dynamics. Moreover, it suggests that 2M background nodes are also essential for detecting fraudsters. Furthermore, we conduct extensive experiments on *DGraph*. Observation and experiments demonstrate that *DGraph* is propulsive to advance GAD research and enable in-depth exploration of anomalous nodes.

## 1 Introduction

Graph data widely presents in various domains and conveys abundant information [41]. Dozens of efforts have been devoted to graph-related research, including node classification [2], link prediction [37], and graph property prediction [49], etc. Among them, Graph Anomaly Detection (GAD) has currently become a hot spot due to its practicability and theoretical value [23; 1]. **Anomalies are a number of nodes, edges and graphs that are distinct from the majority [1].** In real-world scenarios, anomalies are widespread, damaging, but difficult to detect. For example, wire fraudsters are typical anomalies in social networks. As the Federal Bureau of Investigation (FBI) reported in 2020[1], wire fraudsters racked up a whopping $1.8 trillion in losses across 2020. The average victim lost nearly $100,000 last year. These fraudsters involving various of scenarios, such as real estate, investment, etc. However, only about 12% to 15% of all cases get reported and only 29% of victims see their funds fully recovered[2]. GAD aims to detect these anomalies by utilizing network structure information and classic anomaly detection approaches [8; 48]. Thus, investigating GAD is beneficial and applicable in the real world. This paper focuses on anomalous node detection for its representativeness in GAD.

Since "anomaly" is a **scenarios-related** concept, narrowing the gap between academia and industry is the primary requirement of GAD datasets. However, due to the rarity of anomalies in real world,

---

[✉] Corresponding author
[1]https://www.ic3.gov/Media/PDF/AnnualReport/2020_IC3Report.pdf
[2]https://money.com/real-estate-wire-fraud-scam-covid-tips/

36th Conference on Neural Information Processing Systems (NeurIPS 2022) Track on Datasets and Benchmarks.

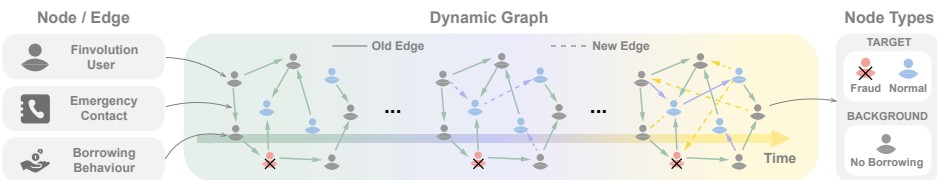

Figure 1: The overview of *DGraph*.

only a small number of public datasets with both graph structure and anomaly ground-truth can be used in GAD research [23], such as Amazon [9], YelpChi [9], and Elliptic [39]. **Thus, enriching the variety of GAD datasets is a fundamental work of current GAD research.** Collecting dataset from some domains that are representative but not covered by current works can greatly speed up this process. Specifically, the financial fraudster detection [40] is such a typical domain. **Meanwhile, current GAD datasets have different limitations, which may gap the current GAD research and practical applications.** Firstly, temporal dynamics of graphs are ignored by most of the current GAD datasets, despite they are being common in the real world [4]. Secondly, scales of current GAD datasets have a gap with industrial scenarios (with more than 1 million nodes) [14]. For example, current GAD datasets which are commonly-used only have 11,944 to 203,769 nodes. Last but not least, in most real-world scenarios, not all the nodes in a graph are actually required to be classified/predicted. But removing these nodes can lose their abundant information and damage the connectivity of network structures, which is somehow like removing background knowledge from a complete story. Therefore, we term these nodes as background nodes and the opposite of them as target nodes. However, most of the current GAD datasets ignore background nodes.

**To enrich the variety of current GAD datasets and overcome their limitations**, we propose *DGraph*, a real-world and large-scale dynamic graph consisting of over 3M nodes and 4M edges. *DGraph* is provided by *Finvolution Group*. It represents a real-world social network in the financial industry. A node represents a *Finvolution* user, and an edge from one user to another means that the user regards the other one as the emergency contact. Besides, the anomalous node in *DGraph* has a practical meaning: the user who has fraudulent behaviors. *DGraph* provides over 1M extremely unbalanced ground-truth nodes, offering a great benefit to evaluation and promotion of previous GAD studies. In addition, *DGraph* preserves more than 2M background nodes, referring to users who are not detection targets in lack of borrow behavior. These nodes are real-world instances and can effectively promote understanding of background nodes in social networks. Meanwhile, *DGraph* contains abundant dynamic information which can be utilized for accurate fraudster identification and further exploration of GAD research. An illustrative overview of the dataset is shown in Fig. 1.

We carefully observe *DGraph* and conduct extensive experiments. The results demonstrate that *DGraph* possesses a variety of novel and promising properties. Firstly, observations suggest that anomalous and normal users in *DGraph* have various characteristics in terms of network structure, the distribution of neighbors' features, and temporal dynamics. **Comprehensively modeling the abundant information of *DGraph* is still a challenge for GAD research**. Besides, observations also demonstrate that background nodes in *DGraph* are vital for detecting fraudsters. ***DGraph* can support and promote the exploration of background nodes in depth.** Last but not least, experiment results of 9 popular supervised and 7 unsupervised methods on *DGraph* reveal that the generalization of current GAD methods is limited. ***DGraph* can offer exciting opportunities to advance previous GAD methods.**

In summary, our contributions are as follows:

- We propose *DGraph*, a real-world and large-scale dynamic graph from financial scenarios.
- We provide a comprehensive observation of *DGraph*, which thoroughly explores the novel and promising properties of *DGraph*.
- We conduct extensive experiments on *DGraph*. The results demonstrate that *DGraph* offers exciting opportunities to advance previous GAD methods.

Our dataset can be found at: https://dgraph.xinye.com/.

## 2   Related datasets

Since graph data is widespread, many works have been devoted to graph research[41; 43]. With the development of graph research, various benchmark datasets are proposed to support and promote the research. These benchmark datasets link to many graph-related tasks, such as node classification [46], link prediction [3] and graph property prediction [12]. Graph anomaly detection (GAD) has currently become a hot research direction due to its practicability and theoretical value. Many efforts go to this topic, aiming to extend GAD to a range of application scenarios [8; 48; 9; 15]. For example, detecting fraudsters on financial platforms [21; 28], anti-money laundering in Bitcoin [22], fake news filtering on social media [9], etc. However, ground-truth anomalies are hard to be collected because of their rarity. Therefore, only Enron [30], Twitter Sybil [10], Disney [32], Amazon [9], Elliptic [39] and YelpChi [9] have both anomaly ground truth and graph structures to date [23]. However, more than half of them are not suited for node-level GAD due to their small-scale network structure. For example, Enron is an email communications dataset, but it has about only 150 users [30]. Therefore, most of the anomalous node detection methods use Amazon, YelpChi, and Elliptic, to evaluate performance. Amazon is constructed from a review dataset provided by *Amazon.com* [24]. Its anomalies are reviews with low ratings. YelphChi is constructed from a review dataset provided by *Yelp.com* [29]. It is worth noting that the label of YelpChi is not the real ground truth since it is constructed by a Yelp Review Filter with about 90% accuracy [26]. Besides, Elliptic is a Bitcoin transaction network provided by *Elliptic.com*, consisting of 203,769 nodes. We provide a detailed summary of these datasets in Table 1.

## 3   Proposed Dataset: *DGraph*

*DGraph* is a dynamic graph that is derived from a real-world finance scenario and linked to a practical application: fraudsters detection. This section introduces the background of raw data in *DGraph* first. Next, we detail the *DGraph* construction process based on raw data. Last but not least, we present the online leaderboard of *DGraph* that is used to track current advancements.

### 3.1   Raw data

The raw data of *DGraph* is provided by *Finvolution Group*[3], a pioneer in China's online consumer finance industry which has more than 140 million registered consumers. *DGraph* focuses on the fintech platform of *Finvolution*, which connects underserved borrowers with financial institutions. According to the financial report of *Finvolution*, more than 14 million consumers borrowed money by this platform during fiscal year 2021, with a total transaction volume of 137.3 billion RMB. The individual borrower must offer a phone number and register an account on *Finvolution* in order to utilize this platform. Users also need to voluntarily complete a basic personal profile, including age, gender, a description of their financial background, etc., which will be used to determine their loan limit. Meanwhile, the emergency contact information is a compulsory requirement. The **emergency contact** is the person of readily available contact information for those in users' life which should be contacted in the event of an emergency. Before commencing each new loan application, users are required to offer at least one contact's name and phone number, which must be kept current. The platform will evaluate loan requests and determine whether or not to give loans to users. In addition, the platform monitors all loans to determine whether users have payed on time and to record the actual repayment date.

The raw data is compiled from the aforementioned information. **It is especially emphasized that all raw data are processed through data masking and strictly respects and protects the privacy of users (see more in Appendix)**. Summarily, the raw data for a specific user includes five components: (1) User id. (2) Basic personal profile information, such as age, gender, etc. (2) Telephone number; note that each account is matched with a specific telephone number. (4) Borrowing behavior, which includes the repayment due date and the actual repayment date. (5) Emergency contacts, which includes the name, telephone number, and last updating time for each contact.

---

[3] https://ir.finvgroup.com/

## 3.2 Graph construction

*Finvolution* has several fraudulent users who cause a significant financial loss for the platform. These fraudsters borrowed money but did not pay it back (far past due), ignoring the platform's repeated reminders. According to common sense and literature [9; 45], fraudsters have some common characteristics which are different from the majority. Therefore, these fraudsters are typical anomalies in *Finvolution* users, to use the common term [1]. And detecting these fraudsters (anomalies) is a typical abnormal detection task as well as an industrial challenge. Financial fraudsters frequently offer false personal information, some of them may also have strange social networks (compared to regular users), and some of them behave abnormally as platform operators. Users' basic profiles are an important component of personal information in raw data that can be used to detect fraudsters. Besides, the emergency contact, which can be treated as a special connection of users, also has some correlation with fraudsters. For one thing, an emergency contact is an important part of a personal profile required by the platform, which can reflect the authenticity of the information provided by users. For another, if users fill true emergency contacts, then this network can be treated as a subgraph of the real-world social network, which can reflect a part of users' social structure. In addition, the filling time of the emergency contact may reveal something about a user's behaviour. Due to these, we construct a GAD dataset called *DGraph* that is built on users' basic profiles and emergency contact links. We also provided a detailed analysis of *DGraph* to verify the correlation of node characteristics and network structure with fraudulent users (see more in Sec. 4).

*DGraph* are constructed in three steps. In the first step, we create *DGraph*'s network structure. We extract users' personal profiles in the second step to build node features. Finally, we label nodes based on their borrowing behaviors. After that, we detail each step of the construction process.

**Step 1. Building the network**. First, we gathered all *Finvolution* users and their corresponding raw data. Next, we select a period of emergency contact records and obtain the user id by matching the telephone number. Then, depending on the user id of the contact, we construct the directed dynamic edge between users, which indicates who is his emergency contact at a given time. In consideration of privacy, we filter some of the emergency contacts, as they are not *Finvolution* users. Then, we construct a graph including all users and edges. From this graph, we take one weakly connected components, which contains **3,700,550 nodes** and **4,300,999 directed edges**, and utilize it as the network structure of *DGraph*. The goal of this operation is to maintain the integrity of the network structure. To safeguard users' privacy, we record the time mark of the edge with a timestamp that can only reflect the time gap between each edge.

**Step 2. Building nodes features**. The node feature derived from the basic personal profile is a vector with 17 dimensions. Each dimension of the node attribute corresponds to a distinct element of the personal profile, such as age and gender. To safeguard the privacy of our users, we do not disclose the significance of any dimension. Since each element of the user's profile is optional (see Sec. 3.1), numerous node attributes miss values. These values are preserved and consistently recorded as "-1", namely, missing values.

**Step 3. Labeling nodes**. 32.2% of the nodes (# 1,225,601) in *DGraph* have related borrowing records. These nodes are labeled based on their borrowing behavior. We define users who exhibit at least one fraud activity, which means they do not repay the loans a long time after the due date and ignore the platform's repeated reminders, as anomalies/fraudsters. Another part of borrowed users are normal users. According to this rule, 15,509 nodes are classified as fraudsters and 1,210,092 nodes as normal users. Except for fraudsters and normal users, *DGraph* comprises 2,474,949 nodes/users (66.8 %) who are registered users but have no borrowing behavior from the platform. These nodes are **background nodes (BN)**. Due to the lack of borrowing behavior, these nodes are not targets for anomaly detection. Nonetheless, these nodes play a crucial role in *DGraph*'s connectivity and it can assist us better identify anomalous nodes (See details in Sec. 4.3). Therefore, they are preserved and labeled as background nodes.

## 3.3 Leaderboard

We provide an online leaderboard for *DGraph*[4], with the goal of assisting researchers in keeping track of current methods and evaluating the efficacy of newly proposed methods. Furthermore, in June

---

[4] https://dgraph.xinye.com/leaderboards/dgraphfin

Table 1: Summary of existing datasets for GAD. In which, "AN" means "Anomalous Nodes", "MV" means "Missing Values", "BN" means "Background Nodes", and "-" means not be reported by the literature. **Note, YelpChi and Amazon*** are re-constructed datasets by [9] based on two reviews dataset: [29] and [24].

| Dataset | # nodes | # edges | # labeled nodes | AN % | MV % | # BN |
|---------|---------|---------|-----------------|------|------|------|
| YelpChi[9] | 45,954 | 3,846,979 | 45,954 | 14.5% | - | 0 |
| Amazon* [9] | 11,944 | 4,398,392 | 11,944 | 9.5% | - | 0 |
| Elliptic[39] | 203,769 | 234,355 | 46,564 | 9.8% | - | 157,205 |
| *DGraph* | **3,700,550** | **4,300,999** | **1,225,601** | **1.3%** | **49.9%** | **2,474,949** |

2022, *Finvolution* will host a deep learning competition[5] based on a dataset that is nearly identical to *DGraph* except for the time involved. *DGraph* and its leaderboard will be used as a competition guide, which will benefit *DGraph*'s promotion. More researchers will be invited to contribute to this exciting new resource.

## 4 Observation on *DGraph*

*DGraph* has a small number of anomalous nodes. Due to the fact that the characteristics of nodes vary in terms of structure, neighbors, and something else, recognizing and interpreting these anomalous nodes is challenging and difficult. In construction, *DGraph* preserves two unique properties: missing values and background nodes. In this section, we make a preliminary observation of this graph, which can help us better comprehend the proposed graph and provide guidance to the question of how to design and interpret models. **In addition, we further explain the results in this section. See more in Appendix.**

### 4.1 Overall

Firstly, we compare *DGraph* with commonly-used graphs in GAD. Table 1 displays a summary of the findings. *DGraph* is the largest public dataset in GAD to date. Specifically, the number of nodes in *DGraph* is **17.1 times** greater than that of Elliptic, with over one million ground-truth and the lowest proportion of anomalies. Therefore, ***DGraph* is a challenging GAD dataset**, requiring a model to process a large number of labeled samples and detect anomalous nodes on samples with the extreme imbalanced classes. It is worth mentioning that *DGraph* is very sparse due to the property of emergency contact. Users will not fill in too many emergency contacts, since the platform do not force users fill as more emergency contact as possible. Therefore, this relationship is naturally sparse. Meanwhile, in construction, we only preserve those users who are *Finvolution* users to protect users' privacy. Thus, many users' emergency contacts who are not *Finvolution* users are filtered, which also leads emergency contact relation become more rare. Besides, Table 1 also show two unique characteristics of *DGraph* . Due to the platform setting (see details in Sec. 3.1), *DGraph* naturally contains 49.9 % missing values. In addition, *DGraph* contains over 2M background nodes, indicating a valuable resource for observing and understanding the function of background nodes in networks.

### 4.2 Anomalous vs. normal

Fraudsters and normal users generally have distinct graph structures and neighbor characteristics. As shown in Fig. 2 (a), fraudsters and normal users have similar average in-degrees, but their average out-degrees differ significantly. The average out-degree of normal users (1.73) is 2.33 times of the fraudsters' (0.75). This result indicates that the graph structure plays a vital role in the detection of fraudsters. Next, we define a neighbor similarity metric in neighbors' features to reveal the similarity between a user's features and its neighbors' feature. The formulation of this metric is $s_i = \mathbf{Avg}(\frac{x_i \cdot x_j}{|x_i||x_j|}|(i,j) \in \mathcal{E})$, where $x_i$ represents the features of node $i$ and $\mathcal{E}$ represents a specific edge set. After that, we group nodes according to their labels and calculate the average neighbor similarity on in- and out-edges for each group. The result is shown in Fig. 2 (b). On average, fraudsters have a lower neighbor similarity than normal users on out-edges, with values of 0.242 and 0.324, respectively. This result suggests that neighbor features also possess an important trait for detecting fraudsters.

---

[5]https://ai.ppdai.com/mirror/goToMirrorDetailSix?mirrorId=28

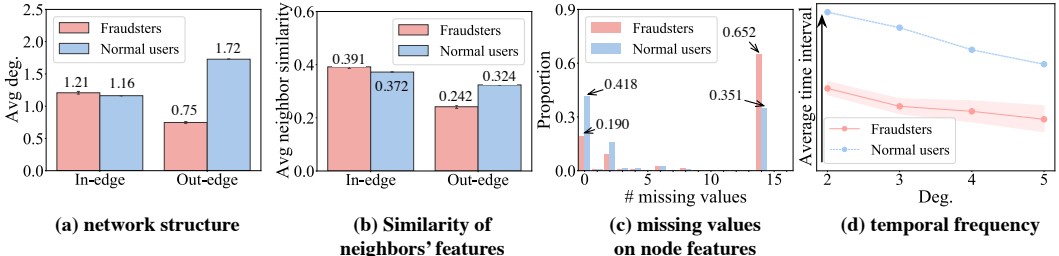

**(a) network structure** **(b) Similarity of neighbors' features** **(c) missing values on node features** **(d) temporal frequency**

Figure 2: Observation of fraudsters and normal users. (a) shows their difference in degrees. (b) shows their difference in neighbors' features. (c) shows their difference in the distribution of missing values. (d) shows their different temporal frequency of edges, and "Average time interval" means the average time interval of node's out-edges.

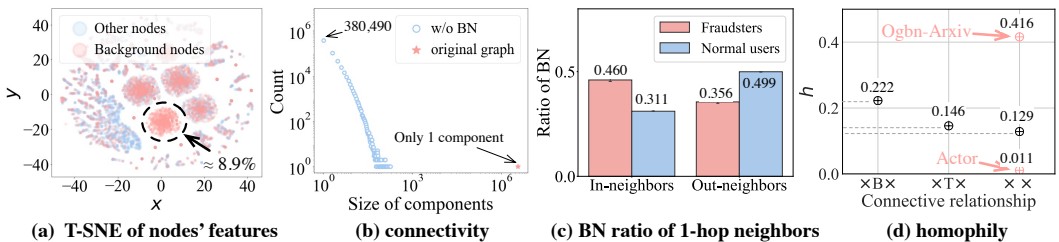

**(a) T-SNE of nodes' features** **(b) connectivity** **(c) BN ratio of 1-hop neighbors** **(d) homophily**

Figure 3: Observation of background nodes. (a) shows that background nodes are hardly separable by node features. (b) illustrates that background nodes are critical for maintaining *DGraph*'s connectivity. (c) shows the average ratio of background nodes in neighbors around nodes. (d) shows homophily ratios of different connective relationships, where "×B×" means two node are connected by a background node, "×T×" means two nodes are connected by a target node, and "××" means two nodes are connected directly.

*DGraph* possesses distinctive characteristics that are also worth investigating in fraudsters identification but usually ignored by many GAD datasets. The existence of missing values and dynamic edge are two particularities of *DGraph*, and they are also helpful in detecting fraudsters. Fig. 2 (c) depicts the proportion of fraudsters and normal users with varying numbers of missing values. As a result of the design of node features, the majority of users have 0 or 14 missing values. Among them, 41.8% of normal users have no missing values, while only 19.0% of fraudsters have no missing values. Consequently, the absence of a value is also a factor that aids in classifying node labels. Meanwhile, *DGraph* provides the last updating date of each edge, allowing us to investigate users' various temporal characteristics. We observe the average time interval of node's out-edges. We group users by their out-degree and count their average interval of out-edges. Fig. 2 (d) demonstrates the result. Overall, higher out-degree nodes have a lower average time interval of out-edges. But fraudsters have a lower average time interval of out-edges than normal users with the same out-degree. This result suggests that fraudsters are more likely to fill their emergency contact information in a short amount of time.

Therefore, in *DGraph*, fraudsters and normal users have differences in aspects of graph structure, neighbor feature distribution, missing values, and temporal dynamics characteristics. In other words, **it can comprehensively be used to evaluate the representational capacity** of graph models.

## 4.3 Background node

The real-world graph is usually massive, redundant, and contains background nodes. For example, in MAG240M [13], only 2 million Arxiv papers are concerned with classification among the 121 million papers. The remaining 119 million papers are not required for the task, but they are useful for node classification due to their significance in maintaining network connectivity and abundance of semantic information. These nodes that are required for classification and prediction are referred to as target nodes, while others are referred to as background nodes. *DGraph* has a lot of background nodes that represent *Finvolution* users who haven't borrowed any money yet, which are ignored by previous GAD datasets. These nodes can assist us in investigating the inherent properties of background nodes.

Although background nodes do not exhibit any borrowing behaviors, there is little distinction between the majority of their features and those of other nodes. We sampled 10,000 target and background nodes and illustrated their characteristics using T-SNE [36]. As shown in Fig. 3 (a), about 92% of background nodes are inseparable from other nodes. Next, nodes are divided into training-set, validation-set, and test-set with a 6/2/2 split setting. Then, we judge whether nodes are background nodes based on their node features using *XGBoost*[6]. The test-set f1-score for the model is only 0.826, which is only a 3.1% improvement over the random guessing with priors (f1-score is 0.801), which means predict all node as background nodes. These results indicate that background nodes are difficult to differentiate based on their features. However, these hardly separable nodes play a crucial role in maintaining the graph's connectivity. Fig. 3 (b) illustrates that the number of weakly connective graph components increases to 605,194, of which 380,490 have a single node after removing background nodes from the *DGraph*. It specifies a vast quantity of target nodes linked by background nodes.

Meanwhile, the background node contains an abundance of semantic information. As shown in Fig. 3 (c), about 46.0% of the in-neighbors of anomalous nodes are background nodes, whereas only 31.1% of the out-neighbors are background nodes. In contrast, the in-neighbors of normal users have a low ratio of background nodes while the out-neighbors have a high ratio of BN. In addition, we observe the role played by the background node in the two-hop relationship. As shown in Fig. 3 (d) We compare the homophily ratio of various connection relationships. we find that 2-hop connection relationship with a background node as intermediate nodes have a higher homophily ratio than others. Moreover, homophily ratios of 2-hop connection relationships are greater than that of two directly connected nodes. Note the reported ratio are measured by the class insensitive edge homophily ratio[19], as well as two popular graphs for comparison: Ogbn-Arxiv[14] and Actor[34]. Therefore, it is worthwhile to investigate how to use background nodes in *DGraph* to enhance performance.

In general, background nodes are essential for maintaining the network's connectivity and contain abundant semantic information for detecting fraudsters. Due to the fact that BN cannot be easily seperated by node characteristics, end2end models rarely use these nodes automatically (see details in Sec. 5). Therefore, **the utilization of the background node merits investigation.**

## 5 Experiments on *DGraph*

*DGraph* is a newly-proposed graph for GAD with an extremely low percentage of anomalous nodes. It possesses a variety of general characteristics. According to the observation, normal users and fraudsters differ in a variety of aspects, such as network structure, temporal dynamics, missing values, and background nodes. In this section, we delve deeper into *DGraph* via extensive experiments, beginning with three questions:

*Q1*: **How powerful are current GAD models on *DGraph*?**

*Q2*: **How to process missing values of *DGraph*?**

*Q3*: **How important are *DGraph*'s background nodes?**

### 5.1 Performance of current models (*Q1*)

**Setup**. We select 9 advanced supervised methods, including 1 baseline methods: MLPs, 4 general graph methods: Node2Vec [11], GCN [17], SAGE [42], and TGAT [42], and 4 anomaly detection methods: DevNet [27], CARE-GNN [9], PC-GNN [21] and AMNet [5]. These methods can capture various graph properties, which are summarized in Appendix. Meanwhile, we select 7 advanced unsupervised methods from PyGOD[6], including: SCAN[44], MLPAE[31], GCNAE[16; 20], Radar[18], DOMINANT[7], GUIDE[47], and OCGNN[38]. We randomly divide the nodes of *DGraph* into training/validation/test sets with a split setting of 70/15/15, respectively. Consider none of the GAD methods we selected can handle directed graph, we convert *DGraph* into an undirected one for comparison. Due to the extreme imbalance of the label distribution, we evaluate models' performance by AUC (ROC-AUC) and AP (Average Precision). See more in Appendix.

**Discussion of supervised methods.** The results of supervised methods are shown in Table 2. First, we observe that MLPs and DevNet that do not utilize any graph information are significantly outper-

---

[6] https://pygod.org/

Table 2: Comparison of AUC and AP achieved by 9 **supervised** methods based on *DGraph*.

| Method | Validation | | Test | |
|---|---|---|---|---|
| | AUC | AP | AUC | AP |
| MLPs | $0.717_{\pm 0.002}$ | $0.026_{\pm 0.000}$ | $0.723_{\pm 0.002}$ | $0.027_{\pm 0.000}$ |
| Node2Vec | $0.626_{\pm 0.002}$ | $0.019_{\pm 0.000}$ | $0.629_{\pm 0.002}$ | $0.020_{\pm 0.000}$ |
| GCN | $0.746_{\pm 0.001}$ | $0.035_{\pm 0.000}$ | $0.751_{\pm 0.002}$ | $0.037_{\pm 0.000}$ |
| SAGE | $0.770_{\pm 0.001}$ | $0.039_{\pm 0.001}$ | $0.778_{\pm 0.001}$ | $0.043_{\pm 0.001}$ |
| TGAT | $\mathbf{0.783}_{\pm 0.001}$ | $\mathbf{0.041}_{\pm 0.000}$ | $\mathbf{0.792}_{\pm 0.001}$ | $\mathbf{0.044}_{\pm 0.001}$ |
| DevNet | $0.707_{\pm 0.001}$ | $0.025_{\pm 0.000}$ | $0.715_{\pm 0.001}$ | $0.026_{\pm 0.000}$ |
| CARE-GNN | $0.734_{\pm 0.004}$ | $0.032_{\pm 0.002}$ | $0.741_{\pm 0.006}$ | $0.033_{\pm 0.002}$ |
| PC-GNN | $0.725_{\pm 0.006}$ | $0.029_{\pm 0.002}$ | $0.734_{\pm 0.006}$ | $0.030_{\pm 0.002}$ |
| AMNet | $0.746_{\pm 0.003}$ | $0.032_{\pm 0.001}$ | $0.752_{\pm 0.003}$ | $0.032_{\pm 0.001}$ |

Table 3: Comparison of AUC and AP achieved by 7 **unsupervised** methods based on *DGraph*. "OOM!" means out-of-memory and "TLE!" means time limit of 24 hours exceeded.

| Method | Validation | | Test | |
|---|---|---|---|---|
| | AUC | AP | AUC | AP |
| SCAN | TLE! | TLE! | TLE! | TLE! |
| MLPAE | $\mathbf{0.625}_{\pm 0.010}$ | $0.017_{\pm 0.001}$ | $\mathbf{0.625}_{\pm 0.011}$ | $0.018_{\pm 0.001}$ |
| GCNAE | $0.497_{\pm 0.003}$ | $0.012_{\pm 0.000}$ | $0.507_{\pm 0.003}$ | $0.013_{\pm 0.000}$ |
| Radar | OOM! | OOM! | OOM! | OOM! |
| DOMINANT | OOM! | OOM! | OOM! | OOM! |
| GUIDE | TLE! | TLE! | TLE! | TLE! |
| OCGNN | $0.616_{\pm 0.004}$ | $\mathbf{0.019}_{\pm 0.000}$ | $0.618_{\pm 0.004}$ | $\mathbf{0.019}_{\pm 0.000}$ |

formed by other baselines that utilize both graph information and node features. But Node2Vec, which only utilizes graph structure, is surpassed by all others. This suggests that both graph information and node features are key factors in detecting fraudsters. It is worth noting that most GAD methods can not outperform the general GNNs. This result is in contrast to the previous result on Amazon and YelpChi, suggesting that previous methods may overfit on current GAD datasets. Therefore, *DGraph* can motivate future works to propose more general models. Among all compared methods, TGAT achieves the state-of-art performance since it can capture the most range of information, including dynamic information, node features and graph information. These results indicates that future GAD methods can take account of more graph properties to make progress.

**Discussion of unsupervised methods.** The results of unspervised methods are shown in Table 3. Overall, *DGraph* is compatible with the unsupervised methods (if they can run). The MLPAE and OCGNN achieve best performance in terms of AUC and AP (0.625 and 0.019, respectively). However, the performance of unsupervised methods still has to be improved in comparison to supervised approaches. It is important to note that many unsupervised GAD methods cannot handle *DGraph* due to memory or time constraints, meaning that some of current unsupervised methods disregard these crucial aspects of industrial application—the time complexity and memory cost.

**In general, *DGraph* have rich and novel properties, which offers exciting opportunities for the improvement of previous GAD methods and benefits future works.**

## 5.2 Missing values in *DGraph* (*Q2*)

According to the observation made in Sec. 4, missing values play a crucial role in detecting anomalous nodes. The next problem is how to handle these missing values. Since the treatment of missing values in graphs has not yet been broadly discussed by current graph models, and most GAD methods are GNNs based methods, we evaluate whether some commonly used tricks are applicable to GNNs.

**Setup.** We choose 4 settings to handle missing values, namely, *Default*, it is the default setting; it replaces missing values with "-1". *Trick A*: it involves adding flags and replacing missing values with "-1", where the flag is set as "1" or "0" to indicate whether a dimension's value is missing. In other words, if a node's feature is $[null, 3]$, after adding flag, the node's feature will be $[-1, 3, 1, 0]$, where the last two numbers are flags. *Trick B*: it involves adding flags and replacing missing values with "0". *Trick C*: it involves adding flags and imputing missing values by a prediction method: *IterativeImputer*

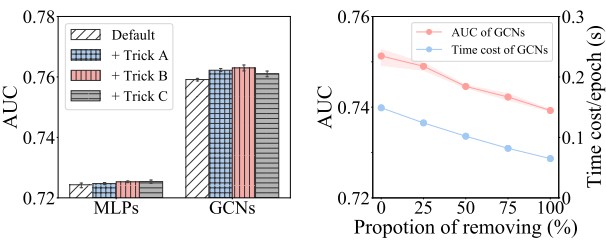

(a) **Handling missing values** (b) **Removing background nodes**

Figure 4: Experiments results. (a) reports results of various tricks of handling missing values. GCNs + *Trick B*, for example, means that we first use *Trick B* to process node features, and then feed the processed features along with the graph structure into GCNs for training and detecting anomalous nodes. (b) reports the decrease of GCNs after we removing different proportion of background nodes. Note that "removing" includes removing both the nodes and their connected edges, and that the percentage range for "removing" is [0, 25, 50, 75, 100] .

[35]. We conduct experiments using MLPs and GCNs whose input node characteristics are processed by these three techniques. See more detailed experiment setup in Appendix.

**Discussion.** The experiment result is shown in Fig. 4 (a). Tricks of handling missing values bring more notable improvements on GCNs than those on MLPs. The average improvement on GCNs is 0.39% of AUC, and that on MLPs is 0.11% of AUC. It suggests that handling missing values for GNNs is indeed necessary. Meanwhile, compared to other tricks on GCNs, *Tricks B* achieves the best improvements. This result indicates that carefully choosing a suitable value for GNNs is also required. However, generally determining a suitable value is complex because the optimal missing value is task-specific. Therefore, how to generally handle missing values on graphs is worth investigating. **DGraph provides an opportunity to explore missing values on the graph**.

## 5.3 Background nodes in *DGraph* (*Q3*)

Background nodes are another distinguishing characteristic of *DGraph*. Observation reveals that background nodes are difficult to differentiate from other nodes but are necessary for maintaining *DGraph*'s connectivity and offering sufficient semantic information for detecting fraudsters. Next, we further investigate how can we utilize background node.

**Removing background nodes**. We remove a variable proportion of background nodes from *DGraph* and feed the remaining graph to GCNs for training and prediction. The experiment setting are identical to those described previously. Fig. 4 (b) shows the result. As the proportion of background nodes being removed increases, the average AUC of GCNs in the test-set decreases from 0.76 to 0.72. These results once again demonstrate the significance of background nodes. It is also worth noting that the time cost of GCNs decreases from 20 to 12 as the proportion of background nodes being removed increases, which indicates a potential direction, which is how to strike a balance between compressing the background nodes to accelerate the model and maintaining the performance.

**Processing background nodes**. According to the observation, background nodes of *DGraph* have abundant semantic information. However, since these nodes and target nodes have tiny differences in the node features, automatically identifying background nodes and utilizing their semantic information is a great challenge for end2end models. Therefore, we conduct an experiment to investigate how can GNNs utilize background nodes. We first add a label indicating whether or not nodes are background nodes into the node features, which is denoted as GCN + *Label*. In addition, we regard the graph as a heterogeneous graph with two types of nodes, the target nodes and the background nodes, and use RGCN [33], a heterogeneous GNNs, to learn the node representation. We restrict the number of RGCN parameters to that of GCN. As shown in Table 4, GCN + *Label* achieves a 2.26% improvement over GCN. Meanwhile, it is surprising that RGCN has a 4.39% improvement over GCN. This result suggests background nodes indeed contains a wealth of semantic information that is ignored by current end2end methods. Therefore, investigating the background nodes is also a promising direction to advance current GAD methods.

**Discussion.** These two experiment results indicate the value of background nodes. **DGraph can be used to explore a general problem: How to process background nodes in any graphs?**

Table 4: Comparison of different methods for processing with background nodes.

| Method | AUC | AP |
|---|---|---|
| GCN | $0.751_{\pm 0.002}$ | $0.037_{\pm 0.000}$ |
| GCN+Label | $0.768_{\pm 0.001}$ | $0.037_{\pm 0.000}$ |
| RGCN | $\mathbf{0.784_{\pm 0.002}}$ | $\mathbf{0.047_{\pm 0.000}}$ |

# 6 Conclusion

This paper presents *DGraph*, a real-world dynamic graph in finance domain, with the aim of enriching the variety of GAD datasets and overcoming the limitations of current datasets. In the construction of *DGraph*, we preserve missing values on node features, and those no-borrowing behaviors nodes are referred to as background nodes. We make a comprehensive observation on *DGraph*. It reveals that anomalous nodes and normal nodes generally have differences on various graph-related characteristic. Meanwhile, the importance of missing values and background nodes is covered by observation. Furthermore, we conduct abundant experiments on *DGraph*, and gain many thought-provoking discoveries. Compared with general GNNs, most current GAD methods present worse performance. It indicates that these GAD methods may overfit on several datasets. Meanwhile, results show that handling missing values and processing background nodes is indeed crucial in *DGraph*. It is expected that these discoveries can be extended to more general fields. In general, *DGraph* overcomes the limitations of current GAD datasets and enriches their varieties. We believe *DGraph* will become an essential resource for a broad range of GAD research.

# 7 Acknowledgments

This work was partially supported by Zhejiang NSF (LR22F020005), the National Key Research and Development Project of China (2018AAA0101900), and the Fundamental Research Funds for the Central Universities.

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
