# OpenReview forum: "DGraph: A Large-Scale Financial Dataset for Graph Anomaly Detection"
_NeurIPS.cc/2022/Track/Datasets_and_Benchmarks — NeurIPS 2022 Datasets and Benchmarks _

### Official Review · Reviewer_MFHZ · 2022-07-13
**A useful dataset but the presentation is rough.**

**Rating:** 6
**Confidence:** 4

**Strengths:**

1. The dataset is the first large-scale graph dataset in graph-based anomaly detection and fraud detection which could help advance this promising research topic. Moreover, the temporal information, missing feature values, and background nodes in this dataset could motivate fruitful and interesting research works in graph machine learning.

2. The preliminary analysis of the dataset is informative and it shares some useful insights about the proposed dataset.

3. The leaderboard and website are well-designed and easy to access.

**Weaknesses:**

1. The writing is rough with multiple grammar errors and typos.

2. The taxonomies used in the paper need to be carefully discussed and defined.

3. I have multiple questions regarding the data collection and the graph construction, please see my detailed comments in the Additional Feedback section.

4. The experiment misses the unsupervised graph anomaly detection method which is a major type of GAD method.

**Additional Feedback:**

1. Grammar errors and typos.
- Line 19, empty reference.
- Line 24, various scenarios.
- Line 59 misses one period after *research*.
- Line 111 misses one period after *privacy*.
- Lines 146-147, repeated sentences.
- Line 173, with the extremely imbalanced classes.
- Line 184, its' -> its.
- Line 189, neighbor features.
- Line 220, training-set.
- experimental result -> experiment result.
- Line 302, conditions -> settings.
- Line 303, testset -> test set.
- Line 322, DGraph can be used to explore a general problem: How to process background nodes in any graphs?
- Line 322 misses one period in the end.



2. Writing suggestions.
- *telecom fraud* can be replaced by *wire fraud* which is more commonly used.
- Only one telecom fraud example cannot support the claim in Line 24 which mentions **various of scenarios**.
- Figure 3, explaining the BN in the caption.
- Line 232, introducing BN before first using it the body text.
- Line 326, the background nodes in the proposed dataset have no borrowing behavior which can be regarded as a type of label and it reflects the neutrality of the user. Therefore, I think the background nodes cannot be called as **unlabeled nodes**.



3. Questions.
- The definition of **fraud** in this dataset is not accurate. A user with only one loan overdue behavior indicates the user's low credibility and high risk. Whether the user is a fraudster (i.e., intentionally missing the payment due) needs more evidence. Thus, I think it is incorrect to describe the anomaly nodes in the dataset as fraudsters. I would like to know the authors' input on this point.

- I am confused about the motivation of each step in graph construction. Why can the demographic information be used to predict one's defaulting behavior? Why do the authors use the **emergency contact** as the relation to build the graph and how does it help identify loan defaulters?

- The proposed dataset is more like a semi-supervised node classification dataset instead of the graph anomaly dataset. Generally, GAD is an unsupervised task but the author did not review and compare the unsupervised GAD methods. If no unsupervised method can identify the anomalies to some extent (e.g., AUC > 60%), it suggests that the node class has a strong correlation with node feature instead of graph semantics. Then I would argue the dataset should be described as a low-credit user detection or node classification dataset in the financial domain. I would like to see the authors' insights on this issue.

- The authors mentioned in Line 105 that emergency contact information is a mandatory requirement for all users which means each user has at least one out-edge. However, Figure 2(a) shows the average out-degree for fraudsters is 0.75, indicating that some fraudsters have no emergency contact. Could authors explain this problem?

- In Line 199, what is the meaning of **cycle**?
- Since the graph is very large, I wonder how the different models are trained on a single GPU?
- In Lines 305-307, the reduced time is inconsistent with the time reported in Figure 4(b). Why is the training time per epoch below 0.3 seconds given such a large graph?





**Clarity:**

The analysis and experiment are generally well-presented but the introduction and dataset curation miss some details and some terms are not used correctly. Please see my detailed comments in the Additional Feedback section.

**Correctness:**

The reason behind choosing node features and connecting users with the emergency contact relationship is not well-explained.

**Documentation:**

The dataset construction part misses some explanations which I will mention in the Additional Feedback section. The authors mentioned multiple times in the paper about protecting the users' privacy during curating the dataset, I would like to see a formal statement (in one section) that describes how the company uses the data and protects users' privacy and whether it complies with privacy legislations like GDPR.

There is no license associated with the proposed dataset.

**Ethics:**

It seems that the published dataset does not contain any personal information. The only concern I have is whether the company asked for users' consent before curating and releasing the dataset.

**Relation To Prior Work:**

The authors give a comprehensive review of existing graph datasets with real-world anomalies. It is better to mention that most previous works only use synthetic graph anomaly datasets to highlight the significant contribution of the proposed real-world graph.

**Summary And Contributions:**

This submission introduces a large-scale dynamic graph dataset from a financial company composed of users as nodes and their social connections as edges. Partial of the users are annotated based on whether they repay the loan in time or not and the overdue users are regarded as anomalies on the graphs. The authors highlight the two uniquenesses of the presented dataset: 1) many nodes are background nodes without labels; 2) many nodes contain missing values in their features. A preliminary analysis of the dataset demonstrates the anomaly nodes differ from the normal nodes from the graph perspective. Experiments using various graph anomaly detectors show the shortcomings of existing methods and also point out the importance of incorporating background nodes and handling missing values in node features. There is a leaderboard and a data competition associated with the dataset which could facilitate future research and practice. Overall, I think the proposed dataset fulfills the urgent need for high-quality datasets in graph-based fraud detection research.

---

> ### Author Response · Authors · 2022-08-28
> **Response to Reviewer MFHZ**
>
> Thank you very much for your thoughtful feedback and suggestions for improvements. We have addressed your concerns based on your comments.
>
> **Thanks for your useful correction of grammar errors and typos.** We have fixed them in the new revision and carefully check the whole paper.
>
> **Your kind suggestion of writing is highly appreciated.** Based on your suggestion, we improved this paper as follows:
> * We have replaced “telecom fraud” with “wire fraud” in the new revision, which indeed is more commonly used.
> * We update a new fraud example in the introduction, which involves more scenarios. Besides, we delete the claim of “various scenarios” in Line 24.
> * In the new revision, we explain “BN” in the caption of Figure 3 and introduce BN in Line 167.
> * And we replaced “unlabeled nodes” in Conclusion with “no-borrowing behavior users”.
>
> **Thanks for your questions about this paper.** Our answers are as follows:
>
> * About **whether the definition of “fraud” is accurate**. Your question reminds us that the description of the fraudulent user is inaccurate in the old version. In fact, the fraudulent user in this paper denotes the user who borrows money but does not pay it back (far past due), ignoring the platform’s repeated reminders. Therefore, these fraudsters are not simply “overdue users”, and they cause a significant financial loss to the platform since the platform will keep informing those overdue users (even their emergency contact) to repay, The platform marks all the users who are overdue for a long time and have subjective fraud behaviors and rejects providing further service for them. DGraph uses the same rules to label nodes, and all nodes are traced for a long time (several months) to get an accurate label. We are sorry for this mistake. We replaced “overdue” with “fraudulent” to avoid ambiguity and updated the related description in Sec.3.2 of the new revision.
>
> * About the **graph construction**. Based on previous studies and common sense, fraudsters usually provide fake personal profiles and some of them may also have irregular social structures (compare to normal users). For the former, an emergency contact is an important part of a personal profile required by the platform, which can reflect the authenticity of the information provided by users. For the latter, if users fill true emergency contacts, then this network can be treated as a subgraph of the real-world social network, which can reflect a part of users’ social structure. Thus, an emergency contact network can help us better infer whether users are fraudsters. We update the above explanations in the new revision, please see Sec. 3.2.
>
> * About the question of **whether we can treat our datasets as anomaly detection datasets.** Firstly, as a general definition of Anomalies, that is “Anomalies are a number of nodes, edges, and graphs that are distinct from the majority ”, the fraudsters in DGraph indeed are such anomalies because the proportion of these users is 3.2%. Thus, we claim DGraph is a GAD dataset. In addition, we supplement the unsupervised baseline in Sec. 5.1, and the experiment results show that two unsupervised algorithms can effectively train and infer fraudsters on DGraph (AUC > 60%).
>
> * About **the node degree in Figure 2**. In construction, we only retain user information of Finvolution users for user privacy protection (See more in Sec. 3.2). Thus, many emergency contacts who are not Finvolution users are filtered, leading to the sparsity of DGraph and some users' degrees become 0.
>
> * About the **"cycle"** in Figure 3(d). We regard users filling a new emergency contact as an event, and "cycle" means an interval of time during which a sequence of a recurring succession of events is completed. Indeed, this definition is confusing for readers. Therefore, we replaced "cycle" with "average interval of out-edges"
>
> * About **the model training**. The dimension of DGraph's node feature is 17, for MLP-based methods, a single GPU can easily handle this scale. For GCN-Based methods, since the edge is very sparse, they can also be fast trained by a single GPU. The only challenge is how to handle the question of “out of memory”. Fortunately, those baselines we report have sampler modules. Therefore, we only utilize a single GPU to search hyper-parameters for each method and get the final results.
>
> * About **GCNs’ training time**. GCN is trained by full-batch strategy. Since the network of DGraph is very sparse and PyG implements automatic parallelization, the training time of GCN is easy to reach 0.3s/epochs.
>
> Besides, **thanks for your concerns in “Ethics” and “Documentation”.** We have updated detailed description regard to these concerns, please see the Appendix of the new revision or see our response to Ethics Reviewer.

---

> ### Comment · Reviewer_MFHZ · 2022-08-29
> **The rebuttal addresses most of my concerns.**
>
> I appreciate the authors' great effort in addressing my concerns and questions. The writing of the revised version has been improved a lot, and the experiment on the unsupervised method is a big add. I have updated my score to 6 accordingly.

---

### Official Review · Reviewer_MofF · 2022-07-15
**This is a good paper about a graph anomaly detection dataset. It proposes a novel GAD dataset with some characteristics. It discusses the necessity and novelty of them.**

**Rating:** 8
**Confidence:** 3
**Correctness:** The dataset is constructed in a sound…

**Strengths:**

1. There is a clear description of how the graph was constructed from real-world data. The nodes, edges, node features, as well as anomalies, are defined clearly meaningful in the real world. One concern when I was reading this paper is that if emergency contacts are reasonable to be edges and construct the graph topology. The concern was solved by the analysis in Figure 2. It shows that normal and abnormal users tend to have different topology structures.
2. It provides clear ablation studies about the missing value and the background nodes. It shows that both of them are necessary designs in GAD and tackling them appropriately improves the performance.
3. Because the data are real-world financial data, the privacy issue is emphasized in the paper.

**Weaknesses:**

When talking about the dynamic graph, Dgraph only uses dynamic edges. Are dynamic nodes also needed in the data? If we have dynamic nodes and their corresponding edges for each timestamp, we have a complete dynamic graph.

**Additional Feedback:**

Thank you for the nice dataset!

**Clarity:**

The paper is well-written and clear. Except for some punctuation marks in lines 19,59, and 111.

**Documentation:**

The website for the dataset is provided. The dataset is easy to download and the README file describes the details clearly.

**Ethics:**

There is a concern about the users' privacy. It is emphasized and anonymized in the paper including the node features and relationships between users. So I do not have further concerns.

**Relation To Prior Work:**

In Table1, DGraph is compared with previous works in size, statistics, and some properties, showing that the authors propose a large-scale GAD dataset with missing values and background nodes that others do not have. These properties are proved to be helpful for GAD in the paper.

**Summary And Contributions:**

This paper introduces the Dgraph dataset, which is a dynamic graph for node anomaly detection. It is a large-scale graph with dynamic edges. The dataset comes from the fintech domain with a clear definition of nodes, edges, node features, and anomalies. Dgraph has the characteristics of missing value and background nodes. These characteristics are proved to be useful when conducting anomaly detection.  Experiments are conducted to show that some current GAD algorithms may overfit several datasets and this makes Dgraph promote the GAD area very possibly.

---

> ### Author Response · Authors · 2022-08-28
> **Response to Reviewer MofF**
>
>
> Sincerely thank you for your appreciation and suggestions to DGraph.
>
> **Thanks for your question about the dynamic of DGraph.** Indeed, in the current version, DGraph only provides edge time, but no node time (e.g., user registration date).
> * First, we would like to explain why we do not publicize node time. In an earlier version of DGraph, we have considered providing a user registration date for releasing, as you say, a complete dynamic graph. In the process of construction, we use linear mapping in time desensitization to keep time information easy to use and understand. However, we found that publicizing the register date and edge time together would bring a huge risk of information leakage.  Specifically,  a person can speculate on our way of encryption by combining the reports of the company and the variety of network sizes, then obtain user behavior data and infer company operational data. Therefore, to avoid the risks of user privacy exposure and negative social impact, we have to delete the node time in the current version.
>
> * For the current version, although real users’ register dates are not available, many methods can utilize the edge time to generate node time. This also has physical meaning. For example, we can define node time as the min time of a node's out-edges. It can approximately represent the earliest activation time of a user on Finvolution, because the user must fill in emergency contact as soon as he or she starts the first loan. Actually, in Table 5, we have already utilized this method to generate node time for TGAT. We update more information about this part in the new revision (See more in Appendix).
>
> Meanwhile, we are actively searching and investigating a special linear encryption method that can encrypt both edge time and node time without user privacy exposure. And we plan to release a complete dynamic graph in a future version.
>
> **Thank you for your correction of writing.** We carefully check typos and update them in the last revision.

---

### Official Review · Reviewer_FwMm · 2022-07-24
**Review of DGraph: A Large-Scale Financial Dataset for Graph Anomaly Detection**

**Rating:** 7
**Confidence:** 4
**Correctness:** The claims made in the submission are…
**Clarity:** The paper is not very well written.

**Strengths:**

The authors present DGraph, a real-world dynamic graph in finance domain, with the aim of enriching the variety of GAD datasets and overcoming the limitations of current datasets. In the construction of DGraph, the authors preserve missing values on node features, and those unlabeled nodes which are referred to as background nodes. The authors make a comprehensive observation on DGraph. It reveals that anomalous nodes and normal nodes generally have differences on various graph-related characteristic. Meanwhile, the importance of missing values and background nodes is covered by observation. Furthermore, the authors conduct abundant experiments on DGraph, and gain many thought-provoking discoveries. Compared with general GNNs, most current GAD methods present worse performance. It indicates that these GAD methods may overfit on several datasets. Meanwhile, results show that handling missing values and processing background nodes is indeed crucial in DGraph. It is expected that these discoveries can be extended to more general fields. In general, DGraph overcomes the limitations of current GAD datasets and enriches their varieties. DGraph could act as an essential resource for a broad range of GAD research.

**Weaknesses:**

The paper is not very well written and needs some further careful elaboration. The proposed graph is very sparse in terms of connections compared with the general graph data. The authors are suggested to explain the sparsity and discuss whether there is edge missing in the data collection.

**Additional Feedback:**

Please refer to strengths and weakness.

**Documentation:**

There is sufficient detail on data collection and organization, availability and maintenance, and ethical and responsible use.

**Ethics:**

The authors claim that they have you read the ethics review guidelines and ensured that your paper conforms to them.

**Relation To Prior Work:**

How this work differs from previous contributions is discussed.

**Summary And Contributions:**

The authors propose DGraph, a real-world and large-scale dynamic graph from financial scenarios. The authors provide a comprehensive observation of DGraph, which comprehensively explores the novel and promising properties of DGraph. The authors conduct extensive experiments on DGraph. The results demonstrate that DGraph offers opportunities to advance previous GAD methods.

---

> ### Author Response · Authors · 2022-08-28
> **Response to Reviewer FwMm**
>
> Thanks for your appreciation and suggestions to DGraph.
>
> **Your correction of writing is highly appreciated.**  We carefully check the typos and inaccurate statements and fixed them in the new revision.
>
> **Thanks for your suggestion of providing an explanation for the sparsity of the graph.** We are sorry to ignore the definition of the emergency contact. Firstly, we will make it clear that an emergency contact list is a list of readily available contact information of those in your life you can ask for help when the event of an emergency occurs. Then, the two reasons for the sparsity of DGraph are presented as follows:
> * In normal circumstances, users will not fill in too many emergency contacts since the platform will not force users to fill as many emergency contacts as possible. Therefore, this relationship is naturally sparse.
> * In the construction process, we only retain user information of  Finvolution users for user privacy protection. Thus, many emergency contacts of those who are not Finvolution users are filtered, leading to the sparsity of DGraph.
>
> We are convinced that the sparsity of DGraph is not caused by edge missing in the data collection. We updated the above explanation in sec 4.1 of the new revision.

---

### Official Review · Reviewer_q3Ce · 2022-07-26
**Lack of important baselines**

**Rating:** 6
**Confidence:** 3
**Correctness:** The dataset is constructed in a sound…

**Strengths:**

- The process of constructing the dataset is elaborated detailedly. The authors provide detailed analysis of anomaly nodes based on the statistics of DGraph, which helps the community profile anomaly nodes in GAD tasks.
- The authors provide amenable website and leaderboard for DGraph, which makes it easy for researchers to use in the future research projects. (All results in this paper could be updated to the leaderboard.)

**Weaknesses:**

- The experiment of GAD baselines is insufficient. More GAD baselines could be added to the experiment. I list some of them as follows. (More related baselines could be found at https://github.com/pygod-team/pygod)

Ding K, Li J, Bhanushali R, et al. Deep anomaly detection on attributed networks[C]//Proceedings of the 2019 SIAM International Conference on Data Mining. Society for Industrial and Applied Mathematics, 2019: 594-602.

Li J, Dani H, Hu X, et al. Radar: Residual Analysis for Anomaly Detection in Attributed Networks[C]//IJCAI. 2017: 2152-2158.

Cai L, Chen Z, Luo C, et al. Structural temporal graph neural networks for anomaly detection in dynamic graphs[C]//Proceedings of the 30th ACM international conference on Information & Knowledge Management. 2021: 3747-3756. (Specifically for dynamic graphs)

- In my opinion, the motivation of this paper is not clear enough. The only strength of DGraph compared with other GAD datasets seems to be its scale. In section 4 and 5, some comparison of DGraph and other datasets which demonstrates the superiority of DGraph could be added. Moreover, the superiority of the temporal information in DGraph could be further discussed.

- Some typos in the paper should be fixed. (1) The citation is missing at line 19. (2) The punctuation is missing at line 59 and line 111. The information source is not clear at footnote 1.

**Additional Feedback:**

Suggestions are provided in the Weaknesses section.

**Clarity:**

Overall, this paper is well organized. In each paragraph, the writing could be improved to make logic clearer.

**Documentation:**

The documentation in this paper is sufficient.

**Ethics:**

In my opinion, there are no potential ethical concerns of this work.

**Relation To Prior Work:**

In my opinion, the improvement of this work from previous works is not fully discussed.

**Summary And Contributions:**

In this paper, the authors propose a large dynamic graph dataset, named DGraph for anomaly detection. Furthermore, they provide a comprehensive observation of DGraph revealing the characteristics of anomaly nodes. Finally, they conduct experiments of GAD (Graph Anomaly Detection) baselines to show the effectiveness of DGraph on this task.

---

> ### Author Response · Authors · 2022-08-28
> **Response to Reviewer q3Ce**
>
> **Thanks for your suggestion of adding more unsupervised GAD baselines.** And your kind recommendation about PyGOD, a useful toolkit for conducting experiments is highly appreciated. Following your suggestion, we have added the results of 7 unsupervised baselines in the new revision. Please see Sec 5.1 in the new revision.
>
> **We are sorry for the ambiguity of our motivation caused by our writing.**  Here we give a clearer statement. Our two motivations are as follows:
> * Since "anomaly" is a scenario-related concept and there are only a few available GAD datasets, supplementing more datasets to enrich the variety of GAD datasets is still a meaningful and fundamental work for community development.
>
> * Meanwhile, currently available GAD datasets respectively have some limitations. The insufficient scale is only a general limitation of them. Apart from this, they always exist several shortages. For example, some of them ignore the dynamic information or the background nodes. These limitations give us an insight that we should collect a new dataset that can overcome current disadvantages.
>
> * Motivated by these two points, we release DGraph, a new and large-scale dataset for prompting GAD research in a better way.
>
> In the new revision, we highlight these three paragraphs to help readers more easily follow this paper.
>
> **Thanks for your correction of typos and footnotes.** We have added a new example with a reachable footnote in Sec.1, fixed typos, and carefully revised the writing of our paper. Please see the new revision.
>
> **Thanks for your suggestion of "clarity"**. Due to the page limit, we had to abbreviate the outline of each section. In the future version, we will extend the outlines of Sec 4 and Sec 5 to offer logically sufficient information for the reader to better follow our paper.

---

> > ### Comment · Reviewer_q3Ce · 2022-08-29
> > **Thank you for your response.**
> >
> > I appreciate your further hard work which makes this paper much better. Your revision addresses my major concerns. While I still think the temporal information deserves more discussion in the experiments as following ways.
> >
> > 1. Only one of the baselines (TGAT) is based on temporal graph. More temporal graph neural network should be added in the main experiments.
> > 2. Temporal information could be added to the ablation study.
> >
> > Overall, I think this is a meaningful benchmark for further research so I would like to raise my score.

---

### Official Review · Reviewer_YKcV · 2022-07-28
**A very good dataset, but the paper needs some improvements**

**Rating:** 5
**Confidence:** 4

**Strengths:**

The paper provides a large-scale real-world graph anomaly detection dataset, which is exactly what researchers and practitioners need in this problem. It can significantly benefit the graph anomaly detection and fraud detection community.

The authors apply both general graph methods and graph anomaly detection methods on DGraph, and provide interesting findings and research directions based on the experiment results.

**Weaknesses:**

In Section 4, although the authors provide some observation of the graph, the analysis is limited. The authors only describe the results but fail to explain the reason and provide some insights into the problem, e.g., why the average out-degree of normal users is higher than fraudsters'? Does it match the domain knowledge?

In experiments, arbitrarily converting DGraph into an undirected graph may affect the results. As the authors discuss in section 4, in-edges and out-edges distribute differently. Retaining the directions of edges may help improve the correctness of the experiments.

**Additional Feedback:**

There are a few places I do not understand. Could you please answer the following question during the discussion?

- In line 223, how to reach 0.801 in the f1 score with a random guess?

- In Fig 3 (d), why 2-hop homophily (XBX/XTX) is compared with 1-hop's (XX)?

**Clarity:**

Frequent minor mistakes in the paper, but the mistakes won't significantly affect the readers' understanding of the paper.

- line 19: missing reference

- line 59, 111: missing dot

- line 101: 137.3 billion RMB

- line 147: duplicate

- fig. 3 (d): missing annotation

- line 322: generally

**Correctness:**

The majority of the claims in the paper are correct, but some contain factual flaws. In Table 1, the Amazon dataset actually should have 3305 background nodes instead of 0.

In line 82, the authors state that more than half of the current GAD datasets are not suited for anomalous node detection due to the network structure. Most of the datasets are released with GAD methods. Network structure is important for detecting anomalies in those datasets.

**Documentation:**

The paper describes how the graph is constructed. Data, descriptions, statistics, and even leaderboards are publicly available on the website. But some of the attributes in the data remain ambiguous, such as edge type and the difference between class 2 and class 3 in the node label.

**Ethics:**

The dataset does not disclose user identity information and the authors protect user privacy by several means, e.g., using time marks instead of timestamps to reflect the time gap between each edge.

**Relation To Prior Work:**

DGraph is an unprecedented large-scale real-world financial graph. The methods used to construct the graph and label the nodes are very solid. The graph has a large portion of missing values and background nodes, which contain abundant information.

**Summary And Contributions:**

The paper provides DGraph, a new large-scale graph anomaly detection dataset containing temporal information. The graph is naturally constructed in a real-world financial scenario. The authors also provide some observations and experiment results on the graph. Finally, a few insights into the graph and potential research directions on the data are pointed out by the authors.

---

> ### Author Response · Authors · 2022-08-28
> **Response to Reviewer YKcV**
>
> Appreciate your thoughtful feedback. We have improved our paper and uploaded a new revision based on your suggestions.
>
> **Thanks for your suggestion about providing more explanation in observation.** Considering the page space is limited, we remove explanations of observation results in the submission. Sorry if this has caused you confusion. The results in observation are consistent with common sense overall. Fraudsters' primary purpose is to defraud from the platform, which motivates them to exhibit a variety of abnormal traits. We can explain the results of the observation (Sec. 3.2) from this point, as follows:
> * Fig. 2 (a). A lower average out-degree indicates that fraudsters tend to fill fewer emergency contacts in general. This phenomenon coincides with their purpose because filling more emergency contacts are helpless in defrauding money.
> * Fig. 2 (b). The emergency contact (EC) relationship is a kind of social connection and the literature suggests that users with social connections are more similar. However, as fraudsters could provide the platform with a false list of emergency contacts to avoid being caught, they may not have social connections with the emergency contacts they filled. As a result, according to [1], the average feature similarity of fraudsters' out-edges is lower than that of normal users.
> * Fig. 2 (c). Since their purpose is to borrow money as soon as possible, fraudsters will not carefully fill out optional items in their personal profile, which will cause their node features to have a large percentage of missing values.
> * Fig. 2 (d). This result suggests that some fraudsters may fill in multiple emergency contacts, but they often fill in their emergency contact within a short time. This is in line with their purpose. Adding more emergency contact are helpless for frauding money.
>
> Note that the background node is a new concept proposed by this paper and they are a unique component of DGraph. This paper only makes a preliminary exploration of them (in Sec. 4.3 and Sec. 5.3) to show their properties (values). We anticipate that subsequent research could focus on the background node and offer novel discoveries. **We have updated the above explanations in Appendix and given a macro explanation in Section 3.2**. Please see the new revision.
>
> **We notice your concern about the experiment setting.** Indeed, observation shows that edges-direction is an important factor for detecting anomalies. However, to the best of my knowledge, no current GAD methods focus on the edge direction. Therefore, we have to uniformly transform DGraph to an undirected one to keep the consistency of general GNNs methods and GAD methods.  We highlight this reason and state the limitation of the current setting in Sec. 5.1 of the new revision.
>
> **Thanks for your concerns about ''correctness''**.
> * The first one is about the background nodes of the Amazon dataset. We are not sure which “Amazon dataset” is your point since the “Amazon dataset” has various versions. In this paper, we emphasize the version of Amazon used by this paper is introduced by [2] (https://github.com/YingtongDou/CARE-GNN). We download this dataset and carefully check the node labels. All nodes have 1/0 labels, therefore this dataset indeed has no natural background nodes. Noted that we focus on such a version of Amazon. Please see the caption of Table 1.
> * Besides, we are sorry for the inaccurate statement of the current GAD datasets in Sec 2. The correct description is that “more than half of the current GAD datasets are not suited for node-level GAD due to their small-scale network structure”.  We have updated the description of related works in the new revision, please see Sec 2.
>
> **Thank you for your correction of writing.** We carefully check the typos and modify them in the last revision.
>
> **As for your questions, we give answers below**:
> * For the question of “random guess”, “random guess” is “the random guessing with priors (f1-score is 0.801), which means predicting all nodes as background nodes.”. We update this in the new revision, please see Sec. 4.3.
> * For the question of Fig 3(d), it should be noted that "XBX", "XTX", and "XX" all represent distinct user relationships. We compare "XX" with "XBX" and "XTX" to show that 2-hop relationships have different properties than 1-hop relationships.
>
> References:
>
> [1] Miller McPherson, Lynn Smith-Lovin, and James M Cook. Birds of a feather: Homophily in social networks. Annual review of sociology, pages 415–444, 2001.
>
> [2] Yingtong Dou, Zhiwei Liu, Li Sun, Yutong Deng, Hao Peng, and Philip S Yu. Enhancing graph neural network-based fraud detectors against camouflaged fraudsters. In Proceedings of the 29th ACM International Conference on Information & Knowledge Management, pages 315–324, 2020.

---

### Official Review · Reviewer_SeLR · 2022-07-28

**Rating:** 7
**Confidence:** 3

**Strengths:**

Many current GAD-related works on dynamic graphs use datasets that are often small in size, and sometimes even add some anomalous nodes manually on the normal graphs, which does not facilitate comparison with other related works. This dataset presents a large-scale dynamic graph dataset and provides a convenient way to download and make comparisons with other GAD work. The paper also conducts experiments and analyses for the case that the labels or attributes of all nodes are not guaranteed to be complete in large-scale datasets.

**Weaknesses:**

The anomaly node proposed in this paper is a user who is overdue for repayment, and the graph structure is constructed based on the user's emergency contact. In the dataset, the relationship between the graph structure and the final detection result is not very intuitive, and more analysis or proof is needed. Also, for an anomaly detection task, users with overdue payments may not be considered anomalous nodes, but as a separate normal category, because their patterns are relatively independent of their information.

**Additional Feedback:**

The current dataset includes only the complete large-scale data. It may be helpful for related research if suitable subgraphs of smaller sizes are available.

**Clarity:**

The overall structure and delivery of the paper are well presented. The paper provides a focused, detailed, and well-developed account of the work performed.

**Correctness:**

The construction process of this dataset is sound. The article gives the credibility analysis of the original data as well as the process and experiments for handling the partial node nature or label loss of the large-scale data.

**Documentation:**

The paper gives access URLs to datasets and leaderboards of related methods, which are convenient for data acquisition and method comparison. The paper also emphasizes that access to raw data does not violate privacy.

**Ethics:**

I do not consider that there are other ethical issues in the dataset requiring additional review.

**Relation To Prior Work:**

The paper provides a good explanation and analysis of its dynamics and large scale, highlighting the advantages of the paper compared to previous work.

**Summary And Contributions:**

This paper presents a new financial-related dataset in the field of graph anomaly detection, which is both dynamic and large-scale. This dataset fills the lack of large-scale datasets for dynamic graph anomaly detection to a certain extent. The article also analyzes some features of the data itself and provides researchers with a convenient way to access the dataset and a uniform benchmark test.

---

> ### Author Response · Authors · 2022-08-28
> **Response to Reviewer SeLR**
>
> Thank you very much for your thoughtful feedback and suggestions for improvements. We have addressed your concerns based on your comments.
>
> **Thanks for pointing out that we did not clearly state the correlation between emergency contact (graph structure) and fraudsters (the final detection results).** We have updated the relative statement in the new revision, as follows:
>
> * Firstly, based on intuition and common sense, fraudsters usually provide fake personal profiles and some of them also may have irregular social structures (compared to normal users). For one thing, an emergency contact is an important part of a personal profile required by the platform, which can reflect the authenticity of the information provided by users. For another, if users fill true emergency contacts, then this network can be treated as a subgraph of the real-world social network, which can reflect a part of users’ social structure. Thus, the emergency contact network can help us better infer whether users are fraudsters.  We have updated these explanations in Sec. 3.2.
>
> * Then, based on the experiment result in Sec 4.1, methods utilizing graph structure have a better performance than those only utilizing node feature, this result demonstrates that emergency contact relationships can be utilized to detect fraudsters.  Relative discussion can be found in Sec. 5.1.
>
> * In addition, we update an ablation study in the Appendix, replacing the original graph structure of DGraph with a different graph structure, e.g., KNN network, only self-loops. And we find using the original graph structure (emergency contact) can get the best performance in the task of detecting abnormality. This result further reflects the original graph structure (emergency contact) has a high correlation with overdue behavior. Please see more in the Appendix of the new revision.
>
>
> **Thank you for your concern about the connection between the definition of “anomalies” and overdue behaviors.**
> * We replaced “overdue users” with fraud users in the new revision. Because these users do not simply have overdue behavior. In fact, these users borrow money but do not pay it back after repeatedly being reminded by the platform. They are typical financial fraudsters. Please see Sec. 3.2.
>
> * Those users with fraud behaviors are rare compared to most common Finvolution users, which are close to the general definition of the anomaly, namely, “Anomalies are a number of nodes, edges, and graphs that are distinct from the majority” [1]. We highlight this content in the new revision, please see Sec. 1 and Sec. 3.2.
>
> * Besides, for the question of whether anomalies are distinct from others in structure or node feature, or graph structure. We further update 7 unsupervised GAD methods in Sec 5. The results show anomalous users and normal users are indeed different in structure and node feature, and they can be detected by unsupervised/supervised anomaly detection methods.
>
>
> **Thank you for your suggestion of providing a suitable subgraph with a small size.** Considering the sparsity of DGraph, directly sampling a sub-graph from the current DGraph may lead to serious information loss. Therefore, we consider releasing a small connection component as DGraph-tiny from the raw data. However, the disclosure of a new batch of data needs to obey the company's data disclosure process and rules, which will take a certain amount of time. We can only promise to disclose the DGraph-tiny on the DGraph official website before October.

---

### Review · Ethics_Reviewer_4ifY · 2022-08-22

**Recommendation:** 2

**Ethics Documentation:**

See above - I am not in a position to evaluate whether the dataset has proper licencing but would encourage the authors to speak to this question in the paper or in the supplementary materials.

**Ethics Review:**

The NeurIPS ethics guidelines state that authors should consider

"1. Consent to use or share the data. Explain whether you have asked the data owner’s permission to use or share data and what the outcome was. Even if you did not receive consent, explain why this might be appropriate from an ethical standpoint. For instance, if the data was collected from a public forum, were its users asked consent to use the data they produced, and if not, why?"

and

"4. Compliance with GDPR and other data-related regulations. For instance, if the authors collect human-derived data, what is the mechanism to guarantee individuals’ right to be forgotten (removed from the dataset)?"

As far as I can tell, these issues are not discussed although the review form suggests that they *are* addressed in section 6. If this refers to "6. Conclusion" then the claim is mistaken. In the appendix I can find no further reference to these questions.

The authors should provide details about these matters before the paper can be accepted for publication.

---

> ### Author Response · Authors · 2022-08-28
> **Response to Ethics Review of Paper319 by Ethics Reviewer 4ifY**
>
> Thanks for your suggestions. We address your concerns as follows:
>
> **Response to your comment *“1. Consent to use or share the data. Explain whether you have asked the data owner’s permission...”*:**
>
> During the data collection stage, the behavior data collected by Finvolution Group and the type of data have been approved by the user. The following is a detailed description of our agreement ("User Privacy Protection Policy"[1], Article 1):
> * “When you start the Paipaidai (Finvolution) loan service, you need to perform real-name verification. We will collect your name, mobile phone number, ID number, ID photo...”.
> * “ When you apply for Paipaidai to evaluate the loan amount, you need to provide your personal information for credit extension. You need to provide the following necessary information: …emergency contact information…”
>
> From an ethical point of view, we follow the GDPR data minimization principle, and all data used by DGraph is necessary for the platform's anti-fraud algorithm.
>
> **Response to your comment *“4. Compliance with GDPR and other data-related regulations. For instance, if the authors collect human-derived data, what is the mechanism to guarantee individuals’ right to be forgotten (removed from the dataset)?”*:**
>
> It is worth noting that DGraph strictly follows the "Personal Information Protection Law of the People's Republic of China"[2] as all the raw data of  DGraph is collected within China. Meanwhile, we also followed GDPR. The data we disclose has equipped with a strict encryption algorithm to ensure that the data is disclosed in an anonymous way. Anonymized data is defined by the "Personal Information Protection Law of the People's Republic of China"[2] at:
> * Article 4, “Personal information refers to various information related to identified or identifiable natural persons and recorded electronically or in other ways, excluding anonymized information.”
> * Article 73, “The meanings of the following terms in this Law: (3) De-identification refers to the process in which personal information is processed so that it cannot identify a specific natural person without the aid of additional information. (4) Anonymization refers to the process in which personal information cannot identify a specific natural person and cannot be recovered after processing.”
>
> In terms of the specific implementation, we anonymize the user by deleting the personal identification and randomizing the user order. The user ID thus cannot be traced back. Moreover, since the user features are not unique, users cannot be identified through the data set, which ensures the anonymity of DGraph.  So, the concerns of the GDPR, such as the correction and deletion requirements from users, will not affect DGraph (as no one can trace or recognize anyone’s data in DGraph, and DGraph does not tie to the individual right). Even so, Finvolution Group still will process data strictly in accordance with the user's data rights.
>
> By following the reviewer’s suggestions, we will continue to track the use of DGraph more closely. Currently, before downloading the data, the user must provide his or her name and email address, and confirm that he or she has read and agreed with the user license, which ensures the non-commercial, unethical, unfair research, or causing negative social impact usage of DGraph (https://dgraph.xinye.com/clause).
>
> **Response to your comment *“As far as I can tell, these issues are not discussed although the review form suggests that they are addressed in section 6. If this refers to "6. Conclusion" then the claim is ...”*:**
>
> We have updated the above content in the Appendix of our paper and fixed the checklist.
>
> References:
>
> [1] User Privacy Protection Policy of Finvolution Group. https://loancontract.ppdai.com/latest/agency/privacy_policy.html
>
> [2] Personal Information Protection Law of the People's Republic of China.
> https://www.npc.gov.cn/npc/c30834/202108/a8c4e3672c74491a80b53a172bb753fe.shtml

---

### Meta-Review · Area_Chair_Dvsj · 2022-09-11

**Recommendation:** Accept
**Confidence:** 4

**Metareview:**

This paper presents a large-scale and real-world dynamic graph in the finance domain, which is applied for anomaly detection. In addition, nine supervised and seven unsupervised methods have been evaluated on DGraph. This paper may push forward the research in the anomaly detection community. However, the presentation of this paper should be further improved. The current version contains many typos and grammar errors (even in the abstract). Moreover, the authors should provide an in-depth analysis of the experimental results and dig deep into the reason behind the empirical observation. In summary, I vote for acceptance of this paper, although would not be upset if it were rejected (i.e., weak acceptance).

---

### Decision · Program_Chairs · 2022-09-16

Accept